# Targeting tumor-associated genes, immune response, and circulating tumor cells in intrahepatic cholangiocarcinoma: Therapeutic potential of *Atractylodes lancea* (Thunb.) DC

Pongsakorn Martviset[ID][1,2], Pathanin Chantree[ID][2,3], Nisit Tongsiri[4], Tullayakorn Plengsuriyakarn[5,6], Kesara Na-Bangchang[ID][5,6]*

1 Division of Parasitology, Department of Preclinical Science, Faculty of Medicine, Thammasat University, Pathum Thani, Thailand, 2 Graduate Program in Applied Biosciences, Faculty of Medicine, Thammasat University, Pathum Thani, Thailand, 3 Division of Anatomy, Department of Preclinical Science, Faculty of Medicine, Thammasat University, Pathum Thani, Thailand, 4 Department of Surgery, Sakon Nakhon Hospital, Sakon Nakhon, Thailand, 5 Graduate Program in Bioclinical Sciences, Chulabhorn International College of Medicine, Thammasat University, Pathum Thani, Thailand, 6 Center of Excellence in Malaria and Cholangiocarcinoma, Thammasat University, Pathum Thani, Thailand

* kesaratmu@yahoo.com

## Abstract

Cholangiocarcinoma (CCA) is one of the most aggressive cancers with a poor prognosis. Current treatment strategies involve hepatobiliary surgery, chemotherapy, radiotherapy, and supportive care; however, the success of these treatments remains limited. Therefore, this study investigated the potential of *Atractylodes lancea* (Thunb) D.C. (AL) in limiting the progress of CCA by targeting the expression of cancer-related genes involved in immune responses and circulating tumor cells. The study was part of Phase 2A clinical trial in advanced-stage intrahepatic iCCA (iCCA) patients: Group 1 (n = 16) received low-dose AL (capsule formulation of the standardized extract of AL: CMC-AL) with standard supportive care, Group 2 (n = 16) received high-dose AL with standard supportive care, and Group 3 (n = 16) received standard supportive care alone. Venous whole blood samples (EDTA, 5 ml) were collected from each patient on Day 1 and Day 90 and the non-CCA subjects (n = 16) on Day 1. Fifty-nine samples (48 and 11 samples for Day 1 and Day 90, respectively) were processed for total RNA isolation. Gene expression was evaluated using reverse transcription followed by a PCR array. Regardless of dosage, gene expression patterns in the AL-treated groups closely resembled those of the healthy subjects. Specifically, cancer-associated genes, including *VEGF-A*, *NR4A3*, *Ki-67*, and *EpCAM*, were significantly down-regulated. Additionally, the expression levels of immune-related genes were modulated in AL-treated patients. The treatment groups exhibited lower levels of the pro-inflammatory cytokine *IL-6*, increased expression of the anti-inflammatory cytokine *IL-10*, and cell-mediated immune-related molecules such as *CTLA4* and

**Data availability statement:** I confirm that the submission contains all raw data required to replicate the results of the study.

**Funding:** This research project was supported by the Thailand Science Research and Innovation Fundamental Fund Fiscal Year 2023 and Thammasat University (Chulabhorn International College of Medicine, Center of Excellence in Pharmacology and Molecular Biology of Malaria and Cholangiocarcinoma). Kesara Na-Bangchang is funded by the National Research Council of Thailand (NRCT), Contract number N42A671041. The funders had no role in study design, data collection and analysis, decision to publish, or preparation of the manuscript.

**Competing interests:** The authors have declared that no competing interests exist.

*PFR1*. These findings suggest the potential of AL for iCCA treatment. However, additional studies are required to confirm the correlation between gene and protein expression profiles, as well as CTCs profile.

## Introduction

Cholangiocarcinoma (CCA), a cancer of the liver's biliary ducts, is one of the most aggressive forms of cancer [1–3]. The development of CCA is associated with several risk factors, including long-term consumption of freshwater fish infected with metacercariae of the liver flukes *Opisthorchis viverrini* and *Clonorchis sinensis*, chronic cholangitis from various causes, and exposure to carcinogens such as *N*-nitrosodimethylamine (NDMA) or dimethylnitrosamine (DMN) [4,5]. CCA is characterized by rapid disease progression and complications, with a generally poor prognosis [6]. The average survival of patients after a definitive diagnosis is typically less than a year [3]. Surgical resection remains the most effective treatment for CCA; however, it is applicable only in a limited number of cases, and the cure rate is lower than 20% [2,3]. Furthermore, resistance to first-line chemotherapeutic agents used for CCA is another significant limitation [7–9]. There is an urgent need for effective treatments to cure or prolong patient survival and improve their quality of life.

*Atractyloides lancea* (AL) Thunb DC. has been recognized for its broad bioactive properties in treating various diseases [10–12]. It has been tested in several forms against CCA, including pure active constituents, crude extracts using various solvents, or formulated products [13–15]. Promising effects of AL have been observed in CCA *in vitro* and *in vivo* [16]. Recently, a Phase 2A clinical trial was conducted using a capsule formulation of the standardized extract of AL (CMC-AL). The results showed that patients who received the high-dose AL had a significantly higher survival rate than those who received the low-dose AL or supportive care alone [17,18].

Biomarkers are critical tools in oncology, helping predict cancer progression, treatment response, and patient prognosis. Among these, circulating tumor cells (CTCs) and circulating immune cells (CICs) hold promise due to their accessibility *via* non-invasive methods like liquid biopsies [19–21]. CTCs are cancer cells that detach from primary or metastatic tumors and enter the bloodstream. They are involved in cancer metastasis and represent a snapshot of the tumor biology. CICs, such as T cells, natural killer (NK) cells, and myeloid-derived suppressor cells (MDSCs), reflect the host immune system's interaction with the tumor. Combining CTC and CIC analysis offers a comprehensive picture of cancer progression. Studies have shown that CTCs can evade immune attack by interacting with platelets or by expressing immune checkpoint ligands. Simultaneous monitoring of CTCs and immune cell profiles enhances the accuracy of prognostic predictions. For example, combining CTC counts with cell phenotypes provides insights into tumor burden and immune status [20–21].

The current study aimed to explore the relationship between the expression levels of CCA-related genes in CTCs and CICs, and CCA progression in patients with

advanced-stage intrahepatic cholangiocarcinoma (iCCA) who were treated with either low- or high-dose AL compared with control (non-AL treated patients and non-CCA subjects). The PCR array was used to detect the expression levels of these marker genes in circulating tumor cells and immune cells [22,23]. Specifically, the study examined the expression levels of genes related to inflammation (interleukin-6: *IL-6*) and anti-inflammation (interleukin-10: *IL-10*), cell-mediated immune-related molecules (T-lymphocyte antigen 4: *CTLA4* and perforin: *PFR1*), apoptosis (fatty acid synthase: *Fas*), cell proliferation (nuclear receptor 4A3: *NR4A3* and *Ki-67*), angiogenesis (vascular endothelial growth factor A: *VEGF-A*), cancer metastasis (epithelial cell adhesion molecule: *EpCAM*), and oxidative stress (nitric oxide synthase 2: *NOS2*). The results obtained would provide critical information on cancer progression in iCCA patients treated with or without AL, contributing to the future development of targeted therapies. In addition, gene expression analysis using serum samples from iCCA patients instead of tissue biopsy may be used as a practical tool to monitor the progress of iCCA.

## Materials and methods

### Study design and patients

The study is part of a single-center, open-label, randomized controlled, phase 2A dose-finding study of CMC-AL in 48 patients with advanced-stage iCCA [16,18]. The study was conducted at Sakon Nakhon Hospital, Sakon Nakhon Province, Northeast Thailand [18]. The world's highest CCA incidence is reported in this area [1–3]. The clinical trial is registered at the Thai Clinical Trials Registry (Protocol no. TCTR20210129007) (https://www.thaiclinicaltrials.org; WHO ICTRP search), and the ethical approval was obtained from the Ethics Committee for Research in Human Subjects, Sakon Nakhon Hospital (Approval no. 043/2020, 2 February 2021). The study complied with Good Clinical Practice (GCP) guidelines and the Declaration of Helsinki [24]. Patients' eligibility criteria, including study procedures were previously described in detail [16,18]. Patients' information (demographic data, clinical and laboratory data) was accessed on 10 October 2023 (**Table 1**). Written informed consent was obtained from each patient before enrolment.

Patients were randomized to three groups to receive treatment for 90 days as follows (**Fig 1**): Group 1 (n = 16): a daily dose of 1,000 mg CMC-AL for 90 days in conjunction with standard supportive care; Group 2 (n = 16): a daily dose of 1,000 mg CMC-AL for 14 days, followed by 1,500 mg for 14 days, and 2,000 mg for 62 days, in conjunction with standard supportive care; and Group 3 (n = 16): standard supportive care alone. The CMC-AL is a capsule formulation of the standardized extract of AL, consisting of the two main active compounds atctylodin (14%, w/w) and beta-eudesmol (6.4%, w/w) [16,18]. The standard supportive care included the provision of general knowledge on CCA, usage of necessary drugs, particularly analgesic drugs to relieve pain, assessment of quality of life, and follow-up of laboratory parameters related to CCA. Blood samples (5 mL) were collected from each patient in Groups 1, 2 and 3 on Days 1 (before treatment) and 90 (the end of treatment) using venipuncture to verify the gene expression level.

For the control group, blood samples were collected only on Day 1. The samples were collected from non-CCA subjects (10 males and 6 females, aged over 18 years, negative OV infection, and without any type of cancer) obtained from previous studies, which were approved by the Thammasat University Ethics Committee for Human Research (COE no. 011/2564 and 011/2564). Written informed consent was obtained from each subject before enrollment. Subjects' information (demographic data and blood samples for each individual subject) was accessed on 23 February 2024. Blood samples were centrifuged at 9,000–12,000 *xg* for 5 min to separate blood components. The buffy coat containing nucleated cells was separated and stored in a liquid nitrogen tank (-180°C) until analysis.

### Analysis of gene expression profiles

**Total RNA extraction.** Total RNA was extracted from buffy coat samples using the QIAamp RNA Blood Mini Kit (Qiagen, Hilden, Germany). Total RNA concentration was determined using NanoDrop™ 2000/2000c Spectrophotometers (Thermo Scientific, Wilmington, DE, USA). The RNAs (10 μg per sample) were reverse transcribed to complementary DNA (cDNA) using

**Table 1. Demographic, clinical and laboratory data of iCCA patients enrolled in the study. Data are presented as median (interquartile range: IQR) or number (%).**

| Baseline characteristics | n (%) | Group 1 (n = 16) | Group 2 (n = 16) | Group 3 (n = 16) |
|---|---|---|---|---|
| -Mass-forming (MF) type | n (%) | 14 (87.8) | 12 (75) | 14 (87.5) |
| -Periductal infiltrating (PI) type | n (%) | 1 (6.25) | 2 (12.5) | 1 (6.25) |
| -MF + PI type | n (%) | 1 (6.25) | 1 (6.25) | 0 (0) |
| -Other (e.g., intraductal growth type) | n (%) | 0 (0) | 1 (6.25) | 1 (6.25) |
| -Liver | n (%) | 2 (12.5) | 3 (16.66) | 3 (16.66) |
| -Distant lymph nodes | n (%) | 9 (44.44) | 10 (55.55) | 9 (50.00) |
| -Regional lymph nodes | n (%) | 2 (11.11) | 0 (0) | 0 (0) |
| -Omentum adenocarcinoma | n (%) | 1 (6.25) | 0 (0) | 1 (6.25) |
| -Bone | n (%) | 0 (0) | 1 (6.25) | 1 (6.25) |
| -Lung | n (%) | 1 (6.25) | 2 (12.5) | 1 (6.25) |
| -Not found | n (%) | 2 (12.5) | 0 (11.11) | 1 (6.25) |
| -IIIB | n (%) | 0 (0) | 2 (12.5) | 1 (6.25) |
| -Stage IIIC: any T + regional lymph node metastasis (N1) + absence of distant metastasis (MO) | n (%) | 0 (0) | 1 (6.25) | 1 (6.25) |
| -Stage IV: any T + Any N + distant metastasis (M1) Surgery | n (%) | 16 (100) | 13 (81.25) | 14 (87.5) |
| **ECOG Performance Score:** | | | | |
| -0-1 | n (%) | 3 (18.75) | 4 (25) | 3 (18.75) |
| -2 | n (%) | 12 (75) | 12 (75) | 13 (81.25) |
| **Karnofsky Performance Score:** | | | | |
| -70 | n (%) | 5 (31.25) | 1 (6.25) | 4 (25) |
| -80 | n (%) | 11 (68.75) | 10 (62.5) | 11 (68.75) |
| -90 | n (%) | 0 (0) | 4 (25) | 1 (6.25) |
| **WBC count ($10^3/\mu L$)** | IQR | 9.07 (7.02-20.5) | 8.7 (6.01-17.5) | 8.65 (3.42-56.5) |
| **Platelet ($10^3/\mu L$)** | IQR | 363.5 (31.3-490) | 294.5 (99-573) | 280 (17-425) |
| **RBC ($10^6/\mu L$)** | IQR | 4.93 (3.54-5.93) | 4.5 (3.83-5.79) | 3.9 (2.5-5) |
| **Hemoglobin (g/dL)** | IQR | 11.95 (7.8-13.4) | 10.9 (10-13.6) | 8.9 (5-13.4) |
| **Hematocrit (%)** | IQR | 35.4 (24.4-39.3) | 33.9 (30.4-43.1) | 26.5 (15-40) |
| **Neutrophil (%)** | IQR | 70 (38.4-84.5) | 68.95 (59.8-87) | 64.5 (44-74) |
| **Lymphocyte (%)** | IQR | 20.55 (11.5-45.4) | 20.7 (7-31.3) | 23.5 (18-39) |
| **Monocyte (%)** | IQR | 5.3 (2.2-7.8) | 2.9 (2-6) | 6 (5-11) |
| **Creatinine (mg%)** | IQR | 0.92 (0.53-1.08) | 0.82 (0.66-1.98) | 0.99 (0.5-1.6) |
| **BUN (mg%)** | IQR | 13 (9.7-27.7) | 14.56 (7.9-19.8) | 12.5 (0.5-31) |
| **AST (IU/L)** | IQR | 52.7 (49.3-77.0) | 66.7 (19.6-95.3) | 43.6 (18.6-88.4) |
| **ALT (IU/L)** | IQR | 54 (26-98) | 28 (21-45) | 23.5 (11-68) |
| **Direct bilirubin (mg%)** | IQR | 0.3 (0.1-0.7) | 0.2 (0.1-0.4) | 0.25 (0.1-0.3) |
| **Total bilirubin (mg%)** | IQR | 0.75 (0.4-0.9) | 0.65 (0.4-0.9) | 0.75 (0.3-1.0) |
| **Total protien (g%)** | IQR | 7.9 (7.1-8.8) | 8.45 (7.8-9.2) | 7.15 (6.9-7.7) |
| **Albumin (g%)** | IQR | 3.05 (1.8-3.6) | 3.45 (2.4-3.9) | 3.45 (2.9-3.7) |
| **ALP** | IQR | 400.5 (107-1024) | 177 (80-502) | 387 (75-504) |
| **CEA (Unit)** | IQR | 10.15 (1.2-292.1) | 4.75 (1.6-33.4) | 267.45 (34.7-500.2) |
| **CA19–9 (Unit)** | IQR | 946.7 (7.8-27332.2) | 2942.5 (2.8-6264.2) | 795.3 (5.8-1584.7) |

*ALP = alkaline phosphatase, ALT = alanine transaminase, AST = aspartate transaminase, BUN = blood urea nitrogen, CEA = CA19–9 =, ECOG = Eastern Cooperative Oncology Group, RBC = red blood cells, WBC = white blood cells*

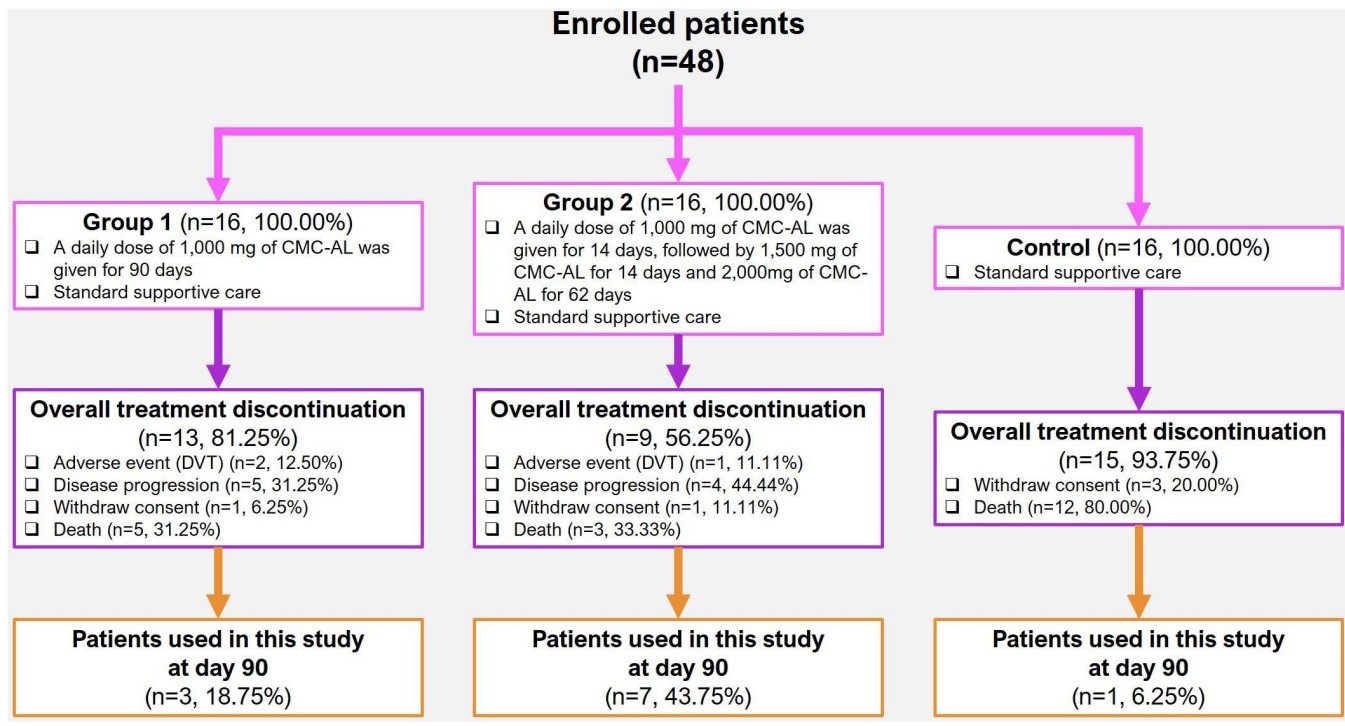

**Fig 1. Study subjects enrolled in this study.**

an RT$^2$ first-strand kit (Qiagen, Hilden, Germany) as per the manufacturer's recommendations. The cDNA concentrations were measured using NanoDrop™ 2000/2000c Spectrophotometers (Thermo Scientific, Wilmington, DE, USA).

**Reverse transcription and PCR array.** The customized RT$^2$ Profiler PCR Array (Qiagen, Germantown, MD, USA) was used to determine the expression levels of target genes in each sample. These target genes included the marker genes for i) inflammation: interleukin-6 (*IL-6*) and interleukin-10 (*IL-10*), ii) apoptosis and inflammation: fatty acid synthase (*Fas*), iii) immune cells: cytotoxic T-lymphocyte-associated protein 4 (*CTLA4*) for T-lymphocytes and perforin (*PFR1*) for NK cells, iv) angiogenesis: vascular endothelium growth factor A (*VEGF-A*), v) cell proliferation: *MKi-67* and nuclear receptor 4 A3 (*NR4A3*), and vi) oxidative stress: nitric oxide synthase 2 (*NOS2*). The human β-actin (*ACTB*) was used as an internal control for each sample, together with the reverse transcription control (*RTC*) and positive PCR control (*PPC*) provided by the kit. The RT$^2$ SYBR® Green ROX™ qPCR Mastermix (Qiagen, Germantown, MD, USA) was used, and the reactions were performed in the StepOnePlus™ Real-Time PCR System (Applied Biosystem, USA). The amplification reaction consisted of the hot start at 95°C for 10 min, followed by 40 cycles of 95°C for 15 sec and 60°C for 1 min. The melting curve was generated between 55–95°C. The PCR array of each sample was performed in triplicate for two independent settings.

## Data analysis

The PCR array results were monitored by real-time PCR, and the gene expression levels were calculated using the StepOne™/StepOnePlus™Software version 2.3 (Applied Biosystem, USA). The Gene Globe software generated the PCR array heatmaps (https://geneglobe.qiagen.com/th/analyze) using maximum correlations at the C$_T$ cut-off of 40. Statistical analysis was performed using SPSS version 28 (IBM, NY, USA). Qualitative data are summarized as the number (%)

of patients available. Quantitative data (gene expression level) are summarized as median (interquartile range). The Kruskal-Wallis test was used to compare the expression levels of each gene across different groups. *Post hoc* pairwise comparisons were performed using the Wilcoxon Signed-Rank test. Multiple testing corrections were applied using PRISM, which computes multiplicity-adjusted *p*-values based on the Bonferroni correction method. The statistical significance level was set at $\alpha = 0.05$ for all tests. All graphs were generated using GraphPad Prism version 10.0 for Windows (GraphPad Software, San Diego, CA, USA).

## Results

A total of 59 samples collected from advanced-stage iCCA patients (n = 16 for each group) and non-CCA subjects on Day 1 were analyzed for the expression levels of all genes. Paired sample analysis (Days 1 and 90) could only be performed in 3 samples (18.75%) from Group 1, 7 samples (43.75%) from Group 2, and 1 sample (6.25%) from Group 3. In Groups 1 and 2 (low- and high-dose CMC-AL), 81.25% (13/16) and 56.25% (9/16) of patients, respectively, discontinued the study due to adverse events such as deep vein thrombosis, disease progression, consent withdrawal, or mortality. In Group 3 (palliative care alone), 93.75% (15/16) of patients died before study completion. **Fig 1** presents detailed gene expression analysis results and survival data across the groups.

### Baseline gene expression levels (Day 1)

The baseline expression levels of each gene in patients across all groups (n = 16 for each group) were compared with those in the non-CCA subjects (**Fig 2**). *IL-10, PFR1, NR4A3*, and *MKi67* expression levels were similar across all groups of iCCA patients and non-CCA subjects. In contrast, the expression levels of *IL-6, CTLP4, Fas, NOS2*, and *EpCAM* were significantly upregulated in iCCA patients compared to non-CCA subjects ($p < 0.005$ for *IL-6, CTLA4, Fas, NOS2*, and *EpCAM*; $p < 0.01$ for *CTLA4*). The expression levels of *VEGF-A* varied among patient groups. In Groups 1 and 2, the expression levels of *VEGF-A* were significantly lower ($p < 0.05$), albeit with only a modest change (< 2-fold).

The expression patterns of nearly all genes in patients from Groups 1, 2, and 3 who completed the study (3, 7, and 1 cases, respectively) were consistent with the overall expression levels observed in all recruited patients. However, individual samples showed different expression levels for specific genes compared to the average for their respective groups. The expression level of *IL-10* in samples from non-CCA subjects was significantly higher (indicated in red) than iCCA patients in all groups. The expression levels of *PFR1, IL-6, NR4A3, MKi67, EpCAM, NOS2, VEGF-A, CTLA4*, and *Fas* in the iCCA groups (Groups 1, 2, and 3) varied widely, ranging from minimal expression (indicated in light green) to maximal expression (indicated in red). In contrast, the expression levels in non-CCA subjects ranged from minimal to average (**Fig 3A**). Group analysis revealed that all genes, except *IL-10*, expression patterns in the iCCA groups were mosaic, while the healthy group showed more consistent, average variations (**Fig 3B**).

### Gene expression levels after treatment (Day 90)

The gene expression patterns in individual patients receiving low-dose (Group 1: No. 1, 2, and 3) and high-dose (Group 2: No. 1, 2, 3, 4, 5, 6, and 7) CMC-AL treatments showed significant changes from baseline levels. In all samples from Groups 1 and 2, the expression levels of *IL-6, Fas, NOS2, VEGF-A, NR4A3, MKi67*, and *EpCAM* were downregulated, while in the Group 3 sample, the expression levels of these genes were upregulated (**Fig 3C**). On the other hand, the expression levels of *IL-10, CTLA4*, and *PFR1* showed variability in Groups 1 and 2. In Group 3, these genes were downregulated (**Fig 3C**). For the overall group analysis, patients in Groups 1 and 2 displayed similar expression patterns, particularly the downregulation of cancer-associated genes (*Fas, NOS2, VEGF-A, NR4A3, MKi67*, and *EpCAM*), as well as the pro-inflammatory cytokine *IL-6*. In contrast, Group 3 patients showed upregulation of all these

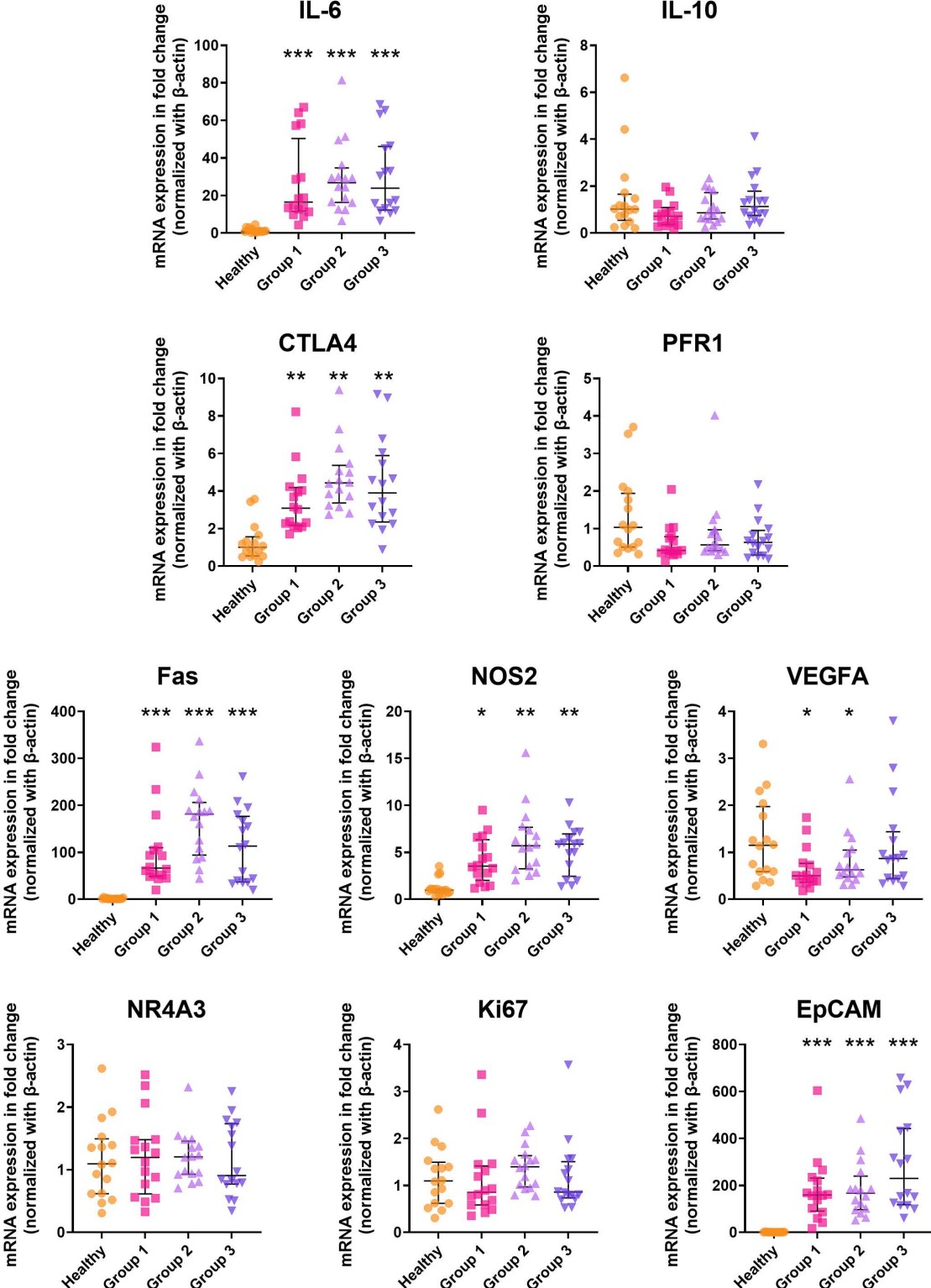

**Fig 2. Baseline of expression levels of the investigated genes in 48 samples collected from iCCA and non-CCA subjects.** Data are presented as individual values, median (horizontal lines) and interquartile range (vertical lines). *Statistically significant difference from non-CCA with *p<0.05, ** p<0.01, and ***p<0.005 (Wilcoxon Signed-Rank test). IL-6: interleukin-6, IL-10: interleukin-10, Fas: fatty acid synthase, CTLA4: cytotoxic*

T-lymphocyte-associated protein 4: PFR1: perforin, VEGF-A: vascular endothelium growth factor A, MKi67: antigen identified by monoclonal antibody Ki-67, NR4A3: nuclear receptor 4 A3, NOS2: nitric oxide synthase 2.

genes. Moreover, genes related to immune response, such as *IL-10*, *CTLA4*, and *PFR1*, were markedly upregulated in Group 2, with varying levels. Only *IL-10* was upregulated in Group 1. In contrast, these immune-related genes were downregulated in Group 3 (**Fig 3D**).

The fold-change analysis obtained from the Gene Globe software revealed upregulation of *IL-6* ($p<0.005$), *Fas* ($p<0.005$), *NOS2* ($p<0.005$), *VEGF-A* ($p<0.005$), *NR4A3* ($p<0.001$), *MKi67* ($p<0.005$), and *EpCAM* ($p<0.005$) in Group 3 patients when compared with non-CCA subjects. In contrast, the expression levels of these genes in Groups 1 and 2 were downregulated to levels similar to those of healthy subjects (**Fig 4**). The expression levels of *IL-10* and *PFR1* varied across groups (**Fig 4**).

## Discussion

The prognosis for patients with CCA remains a significant challenge during treatment due to the aggressive progression of the cancer. This rapid advancement often results in severe complications and death within a short time after a definitive diagnosis [1]. CTCs and CICs have emerged as pivotal biomarkers with significant predictive and prognostic implications in various types of cancer [19–21,25]. Their presence in peripheral blood offers a non-invasive window into the dynamic landscape of cancer progression and treatment response. These insights are crucial for enhancing patient management and developing more effective treatment plans, particularly in advanced or metastatic stages. The buffy coat containing nucleated cells was identified as the most suitable sample for analyzing both CTCs and CICs [22,26,27]. CTCs and CICs have been reported to be potential markers for the early detection of several types of cancer, including CCA, hepatocarcinoma, non-small cell lung cancer, colorectal cancer, squamous cell carcinoma, metastatic head and neck squamous cell carcinoma, pancreatic cancer, and breast cancer [28–36].

The prognosis for patients with CCA remains a significant challenge during treatment due to the aggressive progression of the cancer. This rapid advancement often results in severe complications and death within a short time after a definitive diagnosis [1]. Liquid biopsies, including the detection of circulating tumor cells (CTCs) and circulating invasive cells (CICs), have emerged as promising non-invasive tools for cancer diagnosis, prognosis, and treatment monitoring. Unlike traditional tissue biopsies, liquid biopsies enable real-time tracking of tumor evolution, facilitating early detection of recurrence, treatment resistance, and minimal residual disease. Their minimally invasive nature reduces the risk and discomfort associated with conventional biopsy procedures, making them a valuable approach for longitudinal monitoring in clinical practice [37]. CTCs and CICs have emerged as pivotal biomarkers with significant predictive and prognostic implications in various types of cancer [19–21,25]. Their presence in peripheral blood offers a non-invasive window into the dynamic landscape of cancer progression and treatment response. These insights are crucial for enhancing patient management and developing more effective treatment plans, particularly in advanced or metastatic stages. The buffy coat containing nucleated cells was identified as the most suitable sample for analyzing both CTCs and CICs [22,26,27]. CTCs and CICs have been reported as potential markers for the early detection of several types of cancer, including CCA, hepatocellular carcinoma, non-small cell lung cancer, colorectal cancer, squamous cell carcinoma, metastatic head and neck squamous cell carcinoma, pancreatic cancer, and breast cancer [28–36]. Their ability to reflect tumor heterogeneity and provide insights into molecular characteristics further underscores their clinical relevance in precision oncology.

In CCA, particularly iCCA, the detection of CTCs is becoming increasingly important as a marker for cancer progression and prognosis. CTCs are often present in low numbers in the blood, making detection challenging. CTCs play a crucial role in the early detection of metastasis, monitoring disease progression, and assessing the prognosis of patients with CCA. Although only a limited number of studies have reported on circulating tumor cells in CCA, there is strong evidence supporting the presence of these metastatic cells, particularly in the advanced stages of the disease [23,26,38–40]. CTCs

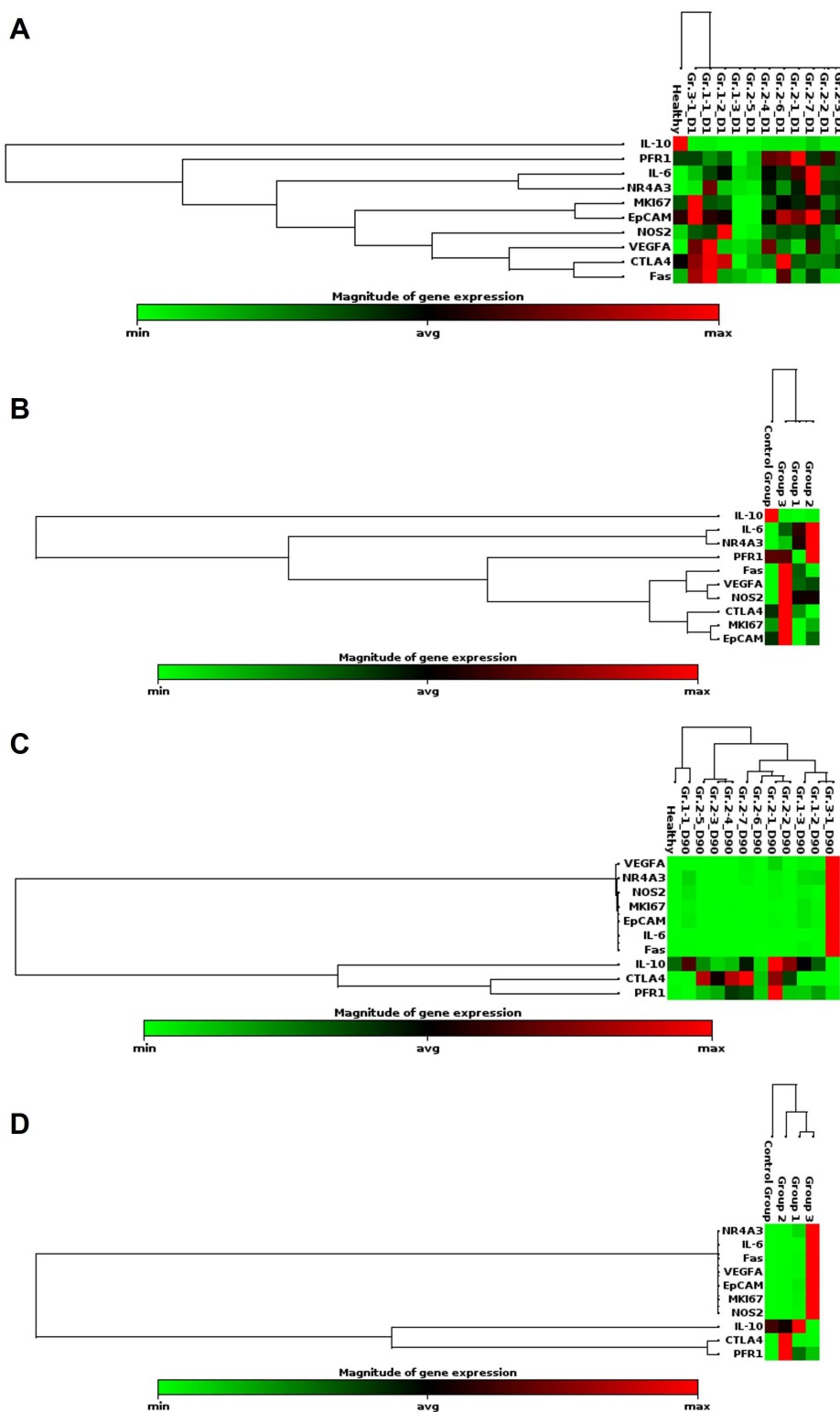

**Fig 3. The clustergrams of the PCR array results: (3A) Day 1 in individual analysis, (3B) Day 1 in group analysis, (3C) Day 90 in individual analysis, and (3D) Day 90 in group analysis.** IL-6: interleukin-6, IL-10: interleukin-10, Fas: fatty acid synthase, CTLA4: cytotoxic T-lymphocyte-associated protein 4: PFR1: perforin, VEGF-A: vascular endothelium growth factor A, MKi67: *antigen identified by monoclonal antibody Ki-67, NR4A3: nuclear receptor 4 A3, NOS2: nitric oxide synthase 2.*

**Fig 4. The analysis of the fold change obtained from the Gene Globe software.** Data are presented as individual values, median (horizontal lines) and interquartile range (vertical lines). *Statistically significant difference from non-CCA with ***p < 0.005 (Wilcoxon Signed-Rank test). IL-6: interleukin-6,*

IL-10: interleukin-10, Fas: fatty acid synthase, CTLA4: cytotoxic T-lymphocyte-associated protein 4: PFR1: perforin, VEGF-A: vascular endothelium growth factor A, MKi67: antigen identified by monoclonal antibody Ki-67, NR4A3: nuclear receptor 4 A3, NOS2: nitric oxide synthase 2.

were detectable in CCA patients and were associated with significantly impaired survival in patients at the metastatic stage. Combined peripheral and central measurements improved the detection rate before surgery to greater than 50% [41]. The prognostic relevance of CTC in CCA patients was described in terms of OS [42].

CCA patients frequently exhibit resistance to chemotherapeutic agents and are often unsuitable candidates for surgical intervention [7]. Consequently, developing novel therapeutic agents and diagnostic tools is crucial and promising [7,43–45]. Our research group has focused on identifying innovative compounds to address these challenges in CCA patients [10,26]. Among the most promising candidates is *A. lancea* (AL) [10]. This plant has been studied extensively, using crude extracts and purified bioactive compounds as standalone treatments or in combination therapies [13,46]. Non-clinical *in vitro* and *in vivo* studies of AL have demonstrated its potential anticancer effects through various mechanisms [11,14,15,47,48]. In the preclinical phase, the safety of the CMC-AL was evaluated in healthy volunteers, with results indicating its safety for further testing [17]. Phase I clinical trials confirmed that CMC-AL modulates immune responses, downregulates pro-inflammatory cytokines, and induces apoptosis in cancer cells [16]. Additionally, its safety profile has been validated through *in vitro, in vivo*, and *in silico* studies, which suggest that CMC-AL can be used safely in humans within an optimized concentration range consistent with pharmacokinetic-pharmacodynamic relationships [18,49]. Based on these findings, the upcoming Phase II clinical trial holds great promise for advancing the treatment of CCA.

In this phase IIA clinical study, upregulated expression of cancer-associated genes, including those linked to tumorigenesis and metastasis, such as *VEGF-A*, *NR4A3*, *MKi67*, and *EpCAM* were detected as early as Day 1 in the iCCA patients (Groups 1, 2, and 3). The representation of the expression of these genes to circulating tumor cells requires confirmation by protein expression and identification of the CTCs. Interestingly, our findings also revealed increased expression of *IL-6* and *CTLA4* in iCCA patients. This may be linked to an enhancement of both humoral and cell-mediated immune responses, which are necessary to combat cancer cells. However, the overexpression of *IL-6*, a key inflammatory mediator, may play a critical role in influencing the prognosis of iCCA patients by exacerbating inflammation [50,51]. Additionally, oxidative stress was observed in iCCA patients, as evidenced by the upregulation of the *NOS2* gene. This is a common feature in cancer patients and is not exclusive to iCCA [52].

Focusing on the eleven samples collected from the patients who remained in the study through Day 90 (3, 7 and 1 from Groups 1, 2, and 3, respectively), the gene expression patterns observed on Day 1 were highly variable and unpredictable. This aligns with a common phenomenon in cancer patients, including those with CCA, where metabolic fluctuations are frequently observed [53–55]. Interestingly, the gene expression of all analyzed genes was downregulated in two volunteers (No. 3 and 5 in Groups 1 and 2, respectively*)*. This might indicate the presence of non-metastatic samples that could not be detected *via* circulating tumor cells. Notably, *IL-10* was the only gene consistently downregulated across all iCCA patients in all groups. This finding is consistent with previous studies, as *IL-10*, an anti-inflammatory cytokine, is typically less expressed in cancer patients [56,57].

The results of the Phase 2A clinical trial suggested that one of the critical indicators of the potential success of CMC-AL in treating iCCA patients was the patients' survival rate [18]. Patients who received CMC-AL, particularly those in the high-dose group (Group 2), exhibited significantly longer survival periods compared to those without CMC-AL treatment (Group 3), as reported in a previous study [18]. Interestingly, analysis of gene expression patterns in Groups 1 and 2 revealed that all cancer-associated genes (*VEGF-A*, *NR4A3*, *MKi67*, and *EpCAM*) were downregulated compared to those in Group 3. Furthermore, oxidative stress markers were also downregulated in Groups 1 and 2 relative to Group 3. These findings suggest the potential of CMC-AL in reducing circulating tumor cells and potentially limiting cancer metastasis.

The cell death marker *Fas* was also downregulated in Groups 1 and 2, indicating a reduction in dead cells in circulation. Immune-related genes also exhibited notable changes. *IL-6* was consistently downregulated in patients exposed to CMC-AL (Groups 1 and 2), while *IL-10*, an anti-inflammatory cytokine, was upregulated compared to the control group (Group 3). Additionally, markers related to cell-mediated immunity showed increased expression levels in CMC-AL-treated groups compared to Group 3. Furthermore, the immune-modulating effects observed in the CMC-AL-treated groups are noteworthy. A significant decrease in the expression of *IL-6* pro-inflammatory cytokine and an increase in the expression of *IL-10* anti-inflammatory cytokine were found. This shift towards an anti-inflammatory response is crucial, as chronic inflammation is a key contributor to cancer progression. Additionally, the upregulation of cell-mediated immune-related molecules such as *CTLA4* and *PFR1* suggests that CMC-AL may enhance the body's immune response to cancer cells. This feature could complement traditional therapies and potentially improve patients' treatment outcomes. The fact that these immune-related changes were observed across both the low- and high-dose groups further emphasizes the potential of CMC-AL to modulate the immune microenvironment in a beneficial manner. This could be particularly relevant in CCA, where immune evasion by tumor cells is a significant challenge. CMC-AL's ability to stimulate both humoral and cell-mediated immunity could help overcome this barrier, offering patients a novel adjunctive treatment option.

Despite these promising results, several limitations should be considered. Gene expression analysis was performed on all nucleated blood cells but not selected CTCs. Additionally, protein expression and CTC analysis were not included in the analysis. Therefore, it is unclear whether any observed changes in gene expression due to circulating immune cells cannot be concluded. The study's relatively small sample size and the fact that only one patient completed the study in the non-AL-treated group (Group 3) may impact the robustness of the findings. The lack of long-term follow-up data limits the ability to conclude whether CMC-AL has sustained effects on survival and quality of life. Further studies with larger patient cohorts, longer follow-up periods, and exploration of the specific molecular mechanisms by which CMC-AL exerts its effects are required to validate these initial findings. Although several cancer-associated genes were downregulated, it remains unclear whether CMC-AL directly affects these pathways or whether the observed changes are part of a broader response to treatment. Investigating the molecular pathways that regulate these genes, including the role of specific phytochemicals in *A. lancea*, could help uncover the precise mechanisms through which CMC-AL impacts tumor progression.

## Conclusion

This study's findings provide valuable insights for developing CMC-AL as a potential treatment for iCCA, particularly in preparation for subsequent clinical trial phases. Based on the analysis of gene expression profiles, our results suggest the potential of *A.lancea* (AL) in modulating inflammation (pro- and anti-inflammatory cytokines) and cell-mediated immune responses. However, additional studies are required to confirm the correlation between gene and protein expression profiles, as well as CTCs profile.

## Supporting information

**S1 File. Table S1-S2 The RT$^2$ profiler PCR array $\Delta C_T$ values (mean±SD) at Days 1 and 90 of the collected samples.** (DOCX)

## Author contributions

**Conceptualization:** Pongsakorn Martviset, Kesara Na-Bangchang.

**Data curation:** Pongsakorn Martviset, Pathanin Chantree, Nisit Tongsiri, Tullayakorn Plengsuriyakarn, Kesara Na-Bangchang.

**Formal analysis:** Pongsakorn Martviset, Pathanin Chantree, Tullayakorn Plengsuriyakarn.

**Funding acquisition:** Kesara Na-Bangchang.

**Investigation:** Pongsakorn Martviset, Nisit Tongsiri.

**Methodology:** Pathanin Chantree.

**Project administration:** Tullayakorn Plengsuriyakarn.

**Resources:** Kesara Na-Bangchang.

**Supervision:** Kesara Na-Bangchang.

**Validation:** Tullayakorn Plengsuriyakarn.

**Visualization:** Kesara Na-Bangchang.

**Writing – original draft:** Pongsakorn Martviset.

**Writing – review & editing:** Pongsakorn Martviset, Kesara Na-Bangchang.

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
