## [Decision Letter · Decision Letter 0]

10 Feb 2025

PONE-D-24-55645Targeting Tumor-Associated Genes, Immune Response, and Circulating Tumor Cells in Intrahepatic Cholangiocarcinoma: Therapeutic Potential of Atractylodes lancea (Thunb.) DC.PLOS ONE

Dear Dr. Na-Bangchang,

Thank you for submitting your manuscript to PLOS ONE. After careful consideration, we feel that it has merit but does not fully meet PLOS ONE’s publication criteria as it currently stands. Therefore, we invite you to submit a revised version of the manuscript that addresses the points raised during the review process.

We look forward to receiving your revised manuscript.

Kind regards,

Keun-Yeong Jeong

Academic Editor

PLOS ONE

**Journal Requirements:**

National Research Council of Thailand.

This research project was supported by the Thailand Science Research and Innovation Fundamental Fund Fiscal Year 2023 and Thammasat University (Chulabhorn International College of Medicine, Center of Excellence in Pharmacology and Molecular Biology of Malaria and Cholangiocarcinoma). Kesara Na-Bangchang is funded by the National Research Council of Thailand (NRCT), Contract number N42A671041.

National Research Council of Thailand.

5. We note that your Data Availability Statement is currently as follows: All relevant data are within the manuscript and its Supporting Information files.

Reviewers' comments:

Reviewer's Responses to Questions

**Comments to the Author**

1. Is the manuscript technically sound, and do the data support the conclusions?

Reviewer #1: Partly

Reviewer #2: No

2. Has the statistical analysis been performed appropriately and rigorously? 

Reviewer #1: Yes

Reviewer #2: No

3. Have the authors made all data underlying the findings in their manuscript fully available?

Reviewer #1: Yes

Reviewer #2: No

4. Is the manuscript presented in an intelligible fashion and written in standard English?

Reviewer #1: Yes

Reviewer #2: Yes

5. Review Comments to the Author

**Reviewer #1: ** Voici la reformulation en anglais :

The presented work is very interesting and deserves to be published. However, certain improvements are necessary to enhance the clarity and rigor of the article. We recommend the following adjustments:

Revision of the quality of the graphs: Improving the resolution and presentation of the graphs would enhance readability and interpretation of the results.

Specification of the extract used: It is essential to detail the chemical composition of the extract studied and specify the location of sample collection. This information will help contextualize the results and ensure scientific reproducibility.

**Reviewer #2:**  Reviewer's Summary: In this exploratory companion study to a phase 2A clinical trial of Atractyloides Lancea in palliative stage Cholangiocarcinoma, Martviset et al. analyze the expression of an array of mRNAs with putative cancer relevance. It is nebulous why the authors chose exactly these genes over other tumor-associated or inflammatory genes. This study has limited use to generate hypotheses, though a non-targeted approach like RNA-seq would've been preferred. However, this study does not sufficiently support the main conclusion, namely that A. lancea reduces circulating tumor cells and inflammation. The methodology is unidimensional and the statistics are questionable in the context of a clinical, i.e., definitive study.

Reviewer's Comments:

Abstract: The abstract is mostly adequate, except for the overdrawn conclusion, as mentioned above. The authors may consider stating more explicitely that all patients were beyond curative treatment intent and in palliative care. It is not immediately apparent that RNA was isolated from all samples and not just from "the non-CCA subjects (n=16) on Day 1..." and this sentence may benefit from splitting in two. In addition, the abstract would benefit from stating the blood fraction which the RNA was isolated from (i.e., the buffy coat).

Competing Interests: Spelling, "The".

Data availability: The declaration and statement on data availability don't fit together. The former states that not all data were available, the latter that all data were included in the manuscript. The latter is false; the authors did not include all raw data. The authors should use this section to explain which data is available and which data is not available (and why). All data that can be made available should be uploaded to a public repository (e.g., JBEI EBB) or included as well-tagged supplementary table. All omissions must be justified.

Line 31: "Blood specimen" could be more specific, e.g., "Heparinized venous whole blood (20ml)" or "central venous whole blood (EDTA, 5ml)".

Line 39 and below: Some gene names are italicized and others are not. The authors may also consider discussing, why they did not analyse protein concentrations of at least some of these markers by multiplex assays like Luminex.

Line 89: I believe the authors are referring to "Marker of proliferation Ki-67" and not "monoclonal antibody Ki-67", given that they measure via PCR. This is a recurring issue throughout the manuscript.

Line 108-109: Baseline clinical and laboratory characteristics of patients in the 4 groups need to be summarized in a table in this clinical trial adjacent study. The superficial summary presented in the text is not sufficient.

Line 122 vs. Line 141: Was informed consent from patients not given in writing? Line 141 specifies "written informed consent" of the control group but only "informed consent" (line 122) from cancer patients. While not necessarily unethical, this may not be lege artis.

Line 143: Are the authors certain they centrifuged at 4000-6000 xG and not at 4000 - 6000 rpm? Such a high acceleration will rupture cells (monocytes and macrophages, for example, start rupturing at >500 xG). Accordingly, this may impair cellular RNA measurements. Buffy coats are usually isolated after 200-500 xG centrifugation over a density separation media, like Ficoll, for 30-45min without break, to isolate this fine cell layer with high purity.

Line 161: CTLA4 not CTLP4 is standard nomenclature. This is a recurring issue throughout the manuscript including in figures.

Line 173 and below, Data analysis: To keep the number of study participants undergoing a non-approved treatment low, clinical studies are regularly definitive. Definitive studies require a priori case number planning. Was a formal case number planning performed? Did the authors account for the expected loss of power due to patient death and loss to follow-up? In addition, the testing of multiple markers (considered independent) likely constitutes multiple testing. The authors do not specify any methods to reduce alpha accumulation in this context, neither for multiple testing between groups (i.e., ANOVA post tests) nor multiple markers (i.e., performing the same tests for IL10, IL6, CTLA4, etc, in the hope that at least one is positive). Normality can almost certainly not be assumed for such small case numbers, independent of any non-normality testing (which is also not specified). Given the 4 group design with two timepoints with missing values, a mixed model approach would probably be most appropriate but will likely require significantly higher case numbers to perform at adequate power. Thus, this companion study must be understood as highly exploratory, and the authors should consider introducing it as such.

Line 178 - 179: "One-way ANOVA..." or "A one-way ANOVA..." may be more appropriate than "The one-way ANOVA...". The same goes for "The paired t-test..." in the next sentence. The authors need to specify which post-test they used after the F-statistic.

Line 190 - 191: "patients [...] discontinued the study [...] due to [...] death." is a quite morbid way of phrasing this sentence, and the authors may consider more sensitive wording.

192: "before study completion."

Line 281: The authors performed RNA extraction on all nucleated blood cells and did not select for CTCs. Established methods include matching DNA mutations to known mutations from the primary tumor or from metastasis biopsies and flow cytometry for tumor cell surface markers, both of which have not been performed. The authors can thus crucially not claim that any of the observed changes in gene expression were due to circulating immune cells. To salvage this, the authors may consider spike-in controls using primary CCA tumor cells or cell lines, to generate a standard curve and estimate CTC counts in their patient samples. A similar overstatement without providing convincing data or linking to their companion study is found in line 308.

Lines 346 - 349: These sentences appear contradictory without further discussion; first the authors note reduced inflammation after AL treatment, then they suggest an improved immune response against he cancer. In addition, the latter is a highly questionable interpretation of upregulated CTLA4, an immunosuppressive molecule. Lastly, as also stated in my comments about the introduction, upregulated RNA can vastly differ from protein expression, which has unfortunately not been analyzed.

Lines 359 - 370: This whole paragraph is almost a word by word repetition of the previous paragraph.

Line 387: Circulating tumor cells have not been directly observed. For such a bold statement, one would expect confirmation by established methods and by at least two methods. Stating that inflammation was reduced would require at a minimum protein measurements, preferably functional tests, like LPS stimulated chemotaxis and ROS production.

Figure legends: It may improve reading flow to also include the chosen statistical test in the figure legends.

Fig. 2: What is presented here? Median or Mean? SEM, SD, 95% CI, IQR? Given the relatively low sample number, Median and Quartiles or Tukey boxplots would probably be most appropriate, while Mean / SD or Mean / 95% CI would also be acceptable due to the additional single point presentation.

Figure 3: I'm not sure if this figure adds enough additional information to warrant a unique figure. The authors may consider omitting this figure entirely, merging it with the previous figure, or moving it to the supplement.

Figure 4: Consider mixed models, as outlined in the comments regarding Data analysis. It is not apparent why the authors chose single point clouds for Fig 2 and violin plots for Fig 4. Violin plots have received a lot of critique lately, mostly pertaining to the regular non-disclosure of basket-size and smoothing function; the presentation in Fig. 2 - given the authors add a disclosure of the measure of central tendency and variance - is thus preferable.

6. PLOS authors have the option to publish the peer review history of their article (what does this mean? ). If published, this will include your full peer review and any attached files.

**Do you want your identity to be public for this peer review?** For information about this choice, including consent withdrawal, please see our Privacy Policy .

Reviewer #1: **Yes: ** mohamed abdelkarim

Reviewer #2: **Yes: ** Willem Buys

---

## [Author Response · Author response to Decision Letter 1]

13 Mar 2025

Response to the Editor to edit the submission:

I confirm that "The funders had no role in study design, data collection and analysis, decision to publish, or preparation of the manuscript."

For the “Funding Statement” section, we do need to provide this detail for all the Funding sources. Otherwise, the published work will not be counted as the output from the funding. The current Funding Statement is required “This research project was supported by the Thailand Science Research and Innovation Fundamental Fund Fiscal Year 2023 and Thammasat University (Chulabhorn International College of Medicine, Center of Excellence in Pharmacology and Molecular Biology of Malaria and Cholangiocarcinoma). Kesara Na-Bangchang is funded by the National Research Council of Thailand (NRCT), Contract number N42A671041.”

With regard to the “Data Availability Statement”, I confirm that the submission contains all raw data required to replicate the results of the study. All relevant data are within the manuscript and its Supporting Information files.

Reviewer #1: Voici la reformulation en anglais :

The presented work is very interesting and deserves to be published. However, certain improvements are necessary to enhance the clarity and rigor of the article. We recommend the following adjustments:

1. Revision of the quality of the graphs: Improving the resolution and presentation of the graphs would enhance the readability and interpretation of the results.

Response: Thank you for your suggestion. Figures 2 and 4 have been revised for clarity. However, Figure 3 was generated from the program; the font size cannot be enlarged.

2. Specification of the extract used: It is essential to detail the chemical composition of the extract studied and specify the location of sample collection. This information will help contextualize the results and ensure scientific reproducibility.

Response: The information on the AL extract used has been added.

Reviewer #2:

Reviewer's Summary: In this exploratory companion study to a phase 2A clinical trial of Atractyloides Lancea in palliative stage Cholangiocarcinoma, Martviset et al. analyze the expression of an array of mRNAs with putative cancer relevance. It is nebulous why the authors chose exactly these genes over other tumor-associated or inflammatory genes. This study has limited use to generate hypotheses, though a non-targeted approach like RNA-seq would've been preferred. However, this study does not sufficiently support the main conclusion, namely that A. lancea reduces circulating tumor cells and inflammation. The methodology is unidimensional and the statistics are questionable in the context of a clinical, i.e., definitive study.

Reviewer's Comments:

1. Abstract: The abstract is mostly adequate, except for the overdrawn conclusion, as mentioned above. The authors may consider stating more explicitly that all patients were beyond curative treatment intent and in palliative care. It is not immediately apparent that RNA was isolated from all samples and not just from "the non-CCA subjects (n=16) on Day 1..." and this sentence may benefit from splitting in two. In addition, the abstract would benefit from stating the blood fraction which the RNA was isolated from (i.e., the buffy coat).

Response: The Abstract has been revised as suggested.

2. Competing Interests: Spelling, "The".

Response: This has been corrected.

3. Data availability: The declaration and statement on data availability don't fit together. The former states that not all data were available, the latter that all data were included in the manuscript. The latter is false; the authors did not include all raw data. The authors should use this section to explain which data is available and which data is not available (and why). All data that can be made available should be uploaded to a public repository (e.g., JBEI EBB) or included as well-tagged supplementary table. All omissions must be justified.

Response: ThePCR array data have been added as the Supplementary information.

4. Line 31: "Blood specimen" could be more specific, e.g., "Heparinized venous whole blood (20ml)" or "central venous whole blood (EDTA, 5ml)".

Response: This has been corrected.

5. Line 39 and below: Some gene names are italicized and others are not. The authors may also consider discussing, why they did not analyse protein concentrations of at least some of these markers by multiplex assays like Luminex.

Response: All genes have been corrected and presented as italic.

6. Line 89: I believe the authors are referring to "Marker of proliferation Ki-67" and not "monoclonal antibody Ki-67", given that they measure via PCR. This is a recurring issue throughout the manuscript.

Response: Thank you for your suggestion. The MKi67 has been changed to Ki67 throughout the manuscript.

7. Line 108-109: Baseline clinical and laboratory characteristics of patients in the 4 groups need to be summarized in a table in this clinical trial adjacent study. The superficial summary presented in the text is not sufficient.

Response: Table 1 with demographic, and baseline laboratory and clinical data of iCCA patients in the three groups have been added.

8. Line 122 vs. Line 141: Was informed consent from patients not given in writing? Line 141 specifies "written informed consent" of the control group but only "informed consent" (line 122) from cancer patients. While not necessarily unethical, this may not be lege artis.

Response: Written informed consents were obtained from all subjects.

9. Line 143: Are the authors certain they centrifuged at 4000-6000 xG and not at 4000 - 6000 rpm? Such a high acceleration will rupture cells (monocytes and macrophages, for example, start rupturing at >500 xG). Accordingly, this may impair cellular RNA measurements. Buffy coats are usually isolated after 200-500 xG centrifugation over a density separation media, like Ficoll, for 30-45min without break, to isolate this fine cell layer with high purity.

Response: This has been corrected as 4,000-6,000 rpm.

10. Line 161: CTLA4 not CTLP4 is standard nomenclature. This is a recurring issue throughout the manuscript including in figures.

Response: CTLP4 has been changed to CTLA4 throughout the manuscript.

11. Line 173 and below, Data analysis: To keep the number of study participants undergoing a non-approved treatment low, clinical studies are regularly definitive. Definitive studies require a priori case number planning. Was a formal case number planning performed? Did the authors account for the expected loss of power due to patient death and loss to follow-up? In addition, the testing of multiple markers (considered independent) likely constitutes multiple testing. The authors do not specify any methods to reduce alpha accumulation in this context, neither for multiple testing between groups (i.e., ANOVA post tests) nor multiple markers (i.e., performing the same tests for IL10, IL6, CTLA4, etc, in the hope that at least one is positive). Normality can almost certainly not be assumed for such small case numbers, independent of any non-normality testing (which is also not specified). Given the 4 group design with two timepoints with missing values, a mixed model approach would probably be most appropriate but will likely require significantly higher case numbers to perform at adequate power. Thus, this companion study must be understood as highly exploratory, and the authors should consider introducing it as such.

Response: All quantitative variables are not normally distributed. Therefore, data are summarized as median (interquartile range). Comparison between groups was performed using the Kruskal Wallis test, followed by paired comparison using the Wilcoxon-Signed Rank test. This part of the statistical analysis has been corrected.

12. Line 178 - 179: "One-way ANOVA..." or "A one-way ANOVA..." may be more appropriate than "The one-way ANOVA...". The same goes for "The paired t-test..." in the next sentence. The authors need to specify which post-test they used after the F-statistic.

Response: One-way ANOVA and paired t-test were not used in the analysis (see above).

13. Line 190 - 191: "patients [...] discontinued the study [...] due to [...] death." is a quite morbid way of phrasing this sentence, and the authors may consider more sensitive wording.

Response: Death is changed to mortality.

14. 192: "before study completion."

Response: This has been corrected.

15. Line 281: The authors performed RNA extraction on all nucleated blood cells and did not select for CTCs. Established methods include matching DNA mutations to known mutations from the primary tumor or from metastasis biopsies and flow cytometry for tumor cell surface markers, both of which have not been performed. The authors can thus crucially not claim that any of the observed changes in gene expression were due to circulating immune cells. To salvage this, the authors may consider spike-in controls using primary CCA tumor cells or cell lines, to generate a standard curve and estimate CTC counts in their patient samples. A similar overstatement without providing convincing data or linking to their companion study is found in line 308.

Response: This limitation has been added and the relevant contents revised.

16. Lines 346 - 349: These sentences appear contradictory without further discussion; first, the authors note reduced inflammation after AL treatment, then they suggest an improved immune response against the cancer. In addition, the latter is a highly questionable interpretation of upregulated CTLA4, an immunosuppressive molecule. Lastly, as also stated in my comments about the introduction, upregulated RNA can vastly differ from protein expression, which has unfortunately not been analyzed.

Response: A significant decrease in the expression of IL-6 pro-inflammatory cytokine and an increase in the expression of IL-10 anti-inflammatory cytokine were found. As inflammation is a key contributor to CCA progression, this may suggest the potential of CMC-AL in controlling CCA progression. Upregulating cell-mediated immune-related molecules such as CTLA4 as well as PFR1 may lead to improvement of body's immune response to cancer cells.

17. Lines 359 - 370: This whole paragraph is almost a word by word repetition of the previous paragraph.

Response: This paragraph has been removed.

18. Line 387: Circulating tumor cells have not been directly observed. For such a bold statement, one would expect confirmation by established methods and by at least two methods. Stating that inflammation was reduced would require at a minimum protein measurements, preferably functional tests, like LPS stimulated chemotaxis and ROS production.

Response: Since the study is considered preliminary and plasma samples were not adequately available for confirmation of protein expression and CTCs analysis. This limitation has been addressed.

19. Figure legends: It may improve reading flow to also include the chosen statistical test in the figure legends.

Response: This has been revised.

20. Fig. 2: What is presented here? Median or Mean? SEM, SD, 95% CI, IQR? Given the relatively low sample number, Median and Quartiles or Tukey boxplots would probably be most appropriate, while Mean / SD or Mean / 95% CI would also be acceptable due to the additional single point presentation.

Response: The analysis of the data presented in Figure 2 has been revised, and the revised figure shows the values with median and interquartile range.

21. Figure 3: I'm not sure if this figure adds enough additional information to warrant a unique figure. The authors may consider omitting this figure entirely, merging it with the previous figure, or moving it to the supplement.

Response: Thank you for your suggestion. This figure provides an overview picture of the PCR array results showing the up- or down-regulated genes from samples collected from the participants who completed the 90-day follow-up period (available paired samples on Day 1 and Day 90) in each group (different number of samples in the four groups). Figure 2 however, shows the baseline expression of all genes at baseline in all groups (n = 16 per group). In addition, the figure presents comparative analysis of gene expression between Day 1 and Day 90.

22. Figure 4: Consider mixed models, as outlined in the comments regarding Data analysis. It is not apparent why the authors chose single point clouds for Fig 2 and violin plots for Fig 4. Violin plots have received a lot of critique lately, mostly pertaining to the regular non-disclosure of basket-size and smoothing function; the presentation in Fig. 2 - given the authors add a disclosure of the measure of central tendency and variance - is thus preferable.

Response: Thank you for your suggestion. Figure 4 has been revised.

---

## [Decision Letter · Decision Letter 1]

30 Mar 2025

PONE-D-24-55645R1Targeting Tumor-Associated Genes, Immune Response, and Circulating Tumor Cells in Intrahepatic Cholangiocarcinoma: Therapeutic Potential of Atractylodes lancea (Thunb.) DC.PLOS ONE

Dear Dr. Na-Bangchang,

Thank you for submitting your manuscript to PLOS ONE. After careful consideration, we feel that it has merit but does not fully meet PLOS ONE’s publication criteria as it currently stands. Therefore, we invite you to submit a revised version of the manuscript that addresses the points raised during the review process.

We look forward to receiving your revised manuscript.

Kind regards,

Keun-Yeong Jeong

Academic Editor

PLOS ONE

Reviewers' comments:

Reviewer's Responses to Questions

**Comments to the Author**

1. If the authors have adequately addressed your comments raised in a previous round of review and you feel that this manuscript is now acceptable for publication, you may indicate that here to bypass the “Comments to the Author” section, enter your conflict of interest statement in the “Confidential to Editor” section, and submit your "Accept" recommendation.

Reviewer #1: All comments have been addressed

Reviewer #2: All comments have been addressed

2. Is the manuscript technically sound, and do the data support the conclusions?

Reviewer #1: Yes

Reviewer #2: Yes

3. Has the statistical analysis been performed appropriately and rigorously? 

Reviewer #1: Yes

Reviewer #2: No

4. Have the authors made all data underlying the findings in their manuscript fully available?

Reviewer #1: Yes

Reviewer #2: Yes

5. Is the manuscript presented in an intelligible fashion and written in standard English?

Reviewer #1: Yes

Reviewer #2: Yes

6. Review Comments to the Author

Reviewer #1: The reviewers have adequately addressed my comments on the manuscript 'Targeting Tumor-Associated Genes, Immune Response, and Circulating Tumor Cells in Intrahepatic Cholangiocarcinoma: Therapeutic Potential of Atractylodes lancea (Thunb.) DC.' and I propose the acceptance of the article.

Reviewer #2: The authors have significiantly reworked the manuscript. The conclusion has been adequately toned down to fit the results. The limitations and further research requirements have been outlined well. It is unfortunate, that a study with such an interesting premise is so strongly impaired by a high loss to follow up from an already small sample size - yet, this is hard to avoid in terminal cancer studies.

I believe that publication is warranted, particularly as this is clinical data, after addressing two major concerns:

- Line 171 - 172: No correction for multiple comparison is detailed here. The local alpha niveau is thus likely inflated to values far higher than 0.05 due to alpha accumulation. Please detail 1) whether your workflow already contained correction for multiple testing, as is the standard in many statistics programs including PRISM, 2) whether you performed multiple testing corrections yourself, such as the Bonferroni-Holm method, or or possibly limited the false discovery rate, or 3) that the study did not correct for alpha accumulation and why.

- Figure 4: I struggle to understand, how samples were selected for this figure, as opposed to all patients in figure 2. As I understand this Figure, only samples were considered, for which data was available on day 1 and 90. However, this does not explain, how the 3 healthy subjects were selected for figure 4, as the methods section specifies they were only sampled once. Figure 4 is central to this study and would thus especially benefit from a clearer description both in the legends, as well as the results section.

Minor comments:

- Line 78-79: This sentence sounds like a citation is missing.

- Line 115-116: "is a capsule formation of"

- Line 131: "enrollment".

- Line 132 - 133: The conversion of RPM to G differs substantially by centrifuge design, i.e., the radius of the rotor. It would be highly beneficial for anyone attempting to reproduce this study, if the acceleration (in xG) was also provided. Alternatively, the authors could provide the exact model of the centrifuge and rotor, so that others can calculate the G-force.

- Line 169: "are summarized as number (%)" I'm not perfectly sure what the authors mean by this, but I assume the authors refer to the number and percent of the population fulfilling or not fulfilling a qualitative criteria.

- "Patients sampled" or "Patients available" may be more appropriate than formulating "Patients used".

- Line 177: The abstract details collection of 59 samples, this sentence mentions 64 samples.

- Table 1: It's great that the authors added this table. For quick visual assessment, the authors may consider also adding "(n,%)" over the first row of data of that table (e.g., second cell, left column). I also believe that they're likely referring to the Karnofsky (not "Kamafsky") Performence Index.

- Line 297-298: The authors don't seem to use this system (Methods), so this sentence my be irrelevant.

- Line 356: Was this Group treated with Placebo? The methods section introduces it as "best supportive care" group.

- 371 - 372: This statement is also made in the sentence prior. As the authors do not analyse immune function as part of this study, this conclusion is likely overdrawn, except if linked to previous work showing functional differences of the immune response.

- Line 375- 376: I appreciate the authors' well written and comprehensive limitation statement. The sentence ending in "...cannot be concluded." is understandable, but doesn't read well. Consider something like "Therefore, it is unclear, whether any observed changes [...] translate to differences in protein expression or cell function."

Additional suggestions:

- Line 282 - 306: The general merit of liquid biopsies and CTC could be explained with much more brevity, to make room for a more thorough discussion of the actual results, as they relate to other studies in the field.

7. PLOS authors have the option to publish the peer review history of their article (what does this mean? ). If published, this will include your full peer review and any attached files.

**Do you want your identity to be public for this peer review?** For information about this choice, including consent withdrawal, please see our Privacy Policy .

Reviewer #1: **Yes: ** Mohamed Abdelkarim

Reviewer #2: **Yes: ** Willem Buys

---

## [Author Response · Author response to Decision Letter 2]

1 Apr 2025

SUMMARY FO RESPONSE TO COMMENTS

PONE-D-24-55645R1

Targeting Tumor-Associated Genes, Immune Response, and Circulating Tumor Cells in Intrahepatic Cholangiocarcinoma: Therapeutic Potential of Atractylodes lancea (Thunb.) DC.

PLOS ONE

Editor:

2. 3. If applicable, we recommend that you deposit your laboratory protocols in protocols.io to enhance the reproducibility of your results. Protocols.io assigns your protocol its own identifier (DOI) so that it can be cited independently in the future.

Comments to the Author

1. If the authors have adequately addressed your comments raised in a previous round of review and you feel that this manuscript is now acceptable for publication, you may indicate that here to bypass the “Comments to the Author” section, enter your conflict of interest statement in the “Confidential to Editor” section, and submit your "Accept" recommendation.

Reviewer #1: All comments have been addressed

Reviewer #2: All comments have been addressed

2. Is the manuscript technically sound, and do the data support the conclusions?

Reviewer #1: Yes

Reviewer #2: Yes

3. Has the statistical analysis been performed appropriately and rigorously?

Reviewer #1: Yes

Reviewer #2: No

4. Have the authors made all data underlying the findings in their manuscript fully available?

Reviewer #1: Yes

Reviewer #2: Yes

5. Is the manuscript presented in an intelligible fashion and written in standard English?

Reviewer #1: Yes

Reviewer #2: Yes

6. Review Comments to the Author

Reviewer #1: The reviewers have adequately addressed my comments on the manuscript 'Targeting Tumor-Associated Genes, Immune Response, and Circulating Tumor Cells in Intrahepatic Cholangiocarcinoma: Therapeutic Potential of Atractylodes lancea (Thunb.) DC.' and I propose the acceptance of the article.

Reviewer #2: The authors have significiantly reworked the manuscript. The conclusion has been adequately toned down to fit the results. The limitations and further research requirements have been outlined well. It is unfortunate, that a study with such an interesting premise is so strongly impaired by a high loss to follow up from an already small sample size - yet, this is hard to avoid in terminal cancer studies.

I believe that publication is warranted, particularly as this is clinical data, after addressing two major concerns:

1) Line 171 - 172: No correction for multiple comparison is detailed here. The local alpha niveau is thus likely inflated to values far higher than 0.05 due to alpha accumulation. Please detail 1) whether your workflow already contained correction for multiple testing, as is the standard in many statistics programs including PRISM, 2) whether you performed multiple testing corrections yourself, such as the Bonferroni-Holm method, or or possibly limited the false discovery rate, or 3) that the study did not correct for alpha accumulation and why.

Response: Thank you for this valuable comment. We have revised the statement in the manuscript: “Multiple testing corrections were applied using PRISM, which computes multiplicity-adjusted p-values based on the Bonferroni correction method.”

2) Figure 4: I struggle to understand, how samples were selected for this figure, as opposed to all patients in figure 2. As I understand this Figure, only samples were considered, for which data was available on day 1 and 90. However, this does not explain, how the 3 healthy subjects were selected for figure 4, as the methods section specifies they were only sampled once. Figure 4 is central to this study and would thus especially benefit from a clearer description both in the legends, as well as the results section.

Response: Thank you for your insightful comment. You are correct that the healthy subjects were sampled only once, as we assumed they would not exhibit significant changes in the evaluated markers over time. For Figure 4, three samples from the healthy group were randomly selected to serve as controls for the PCR array analysis. These results were then used as a reference for comparison in Figure 4. We will clarify this selection process in both the figure legend and the results section to enhance transparency.

Minor comments:

1) Line 78-79: This sentence sounds like a citation is missing.

Response: The references have been added.

2) Line 115-116: "is a capsule formation of"

Response: This has been corrected.

3) Line 131: "enrollment".

Response: This has been corrected.

4) Line 132 - 133: The conversion of RPM to G differs substantially by centrifuge design, i.e., the radius of the rotor. It would be highly beneficial for anyone attempting to reproduce this study, if the acceleration (in xG) was also provided. Alternatively, the authors could provide the exact model of the centrifuge and rotor, so that others can calculate the G-force.

Response: The sentence has been revised as “Blood samples were centrifuged at 9,000-12,000 xg for 5 min to separate blood components.”

5) Line 169: "are summarized as number (%)" I'm not perfectly sure what the authors mean by this, but I assume the authors refer to the number and percent of the population fulfilling or not fulfilling a qualitative criteria.- "Patients sampled" or "Patients available" may be more appropriate than formulating "Patients used".

Response: The sentence has been revised as “Qualitative data are summarized as the number (%) of patients available.”

6) Line 177: The abstract details collection of 59 samples, this sentence mentions 64 samples.

Response: The number has been corrected as 59 samples.

7) Table 1: It's great that the authors added this table. For quick visual assessment, the authors may consider also adding "(n,%)" over the first row of data of that table (e.g., second cell, left column). I also believe that they're likely referring to the Karnofsky (not "Kamafsky") Performence Index.

Response: The column has been added as suggested and the word ‘Karnofsky’ has been corrected.

8) Line 297-298: The authors don't seem to use this system (Methods), so this sentence my be irrelevant.

Response: The sentence has been removed.

9) Line 356: Was this Group treated with Placebo? The methods section introduces it as "best supportive care" group.

Response: This has been corrected as ‘control group’ (supportive treatment alone).

10) 371 - 372: This statement is also made in the sentence prior. As the authors do not analyse immune function as part of this study, this conclusion is likely overdrawn, except if linked to previous work showing functional differences of the immune response.

Response: This statement has been removed.

11) Line 375- 376: I appreciate the authors' well-written and comprehensive limitation statement. The sentence ending in "...cannot be concluded." is understandable, but doesn't read well. Consider something like "Therefore, it is unclear whether any observed changes [...] translate to differences in protein expression or cell function."

Response: The sentence has been revised as “Therefore, it is unclear whether any observed changes in gene expression due to circulating immune cells cannot be concluded.”

Additional suggestions:

12) Line 282 - 306: The general merit of liquid biopsies and CTC could be explained with much more brevity, to make room for a more thorough discussion of the actual results, as they relate to other studies in the field.

Response: This has been revised as “Liquid biopsies, including the detection of circulating tumor cells (CTCs) and circulating invasive cells (CICs), have emerged as promising non-invasive tools for cancer diagnosis, prognosis, and treatment monitoring. Unlike traditional tissue biopsies, liquid biopsies enable real-time tracking of tumor evolution, facilitating early detection of recurrence, treatment resistance, and minimal residual disease. Their minimally invasive nature reduces the risk and discomfort associated with conventional biopsy procedures, making them a valuable approach for longitudinal monitoring in clinical practice [37].”

7. PLOS authors have the option to publish the peer review history of their article (what does this mean?). If published, this will include your full peer review and any attached files.

Do you want your identity to be public for this peer review? For information about this choice, including consent withdrawal, please see our Privacy Policy.

Reviewer #1: Yes: Mohamed Abdelkarim

Reviewer #2: Yes: Willem Buys

---

## [Decision Letter · Decision Letter 2]

15 Apr 2025

Targeting Tumor-Associated Genes, Immune Response, and Circulating Tumor Cells in Intrahepatic Cholangiocarcinoma: Therapeutic Potential of Atractylodes lancea (Thunb.) DC.

PONE-D-24-55645R2

Dear Dr. Na-Bangchang,

We’re pleased to inform you that your manuscript has been judged scientifically suitable for publication and will be formally accepted for publication once it meets all outstanding technical requirements.

Kind regards,

Keun-Yeong Jeong

Academic Editor

PLOS ONE

Reviewers' comments:

Reviewer's Responses to Questions

**Comments to the Author**

1. If the authors have adequately addressed your comments raised in a previous round of review and you feel that this manuscript is now acceptable for publication, you may indicate that here to bypass the “Comments to the Author” section, enter your conflict of interest statement in the “Confidential to Editor” section, and submit your "Accept" recommendation.

Reviewer #1: All comments have been addressed

Reviewer #2: All comments have been addressed

2. Is the manuscript technically sound, and do the data support the conclusions?

Reviewer #1: Yes

Reviewer #2: Yes

3. Has the statistical analysis been performed appropriately and rigorously? 

Reviewer #1: Yes

Reviewer #2: Yes

4. Have the authors made all data underlying the findings in their manuscript fully available?

Reviewer #1: Yes

Reviewer #2: Yes

5. Is the manuscript presented in an intelligible fashion and written in standard English?

Reviewer #1: Yes

Reviewer #2: Yes

6. Review Comments to the Author

Reviewer #1: The reviewers have adequately addressed my comments on the manuscript 'Targeting

Tumor-Associated Genes, Immune Response, and Circulating Tumor Cells in Intrahepatic

Cholangiocarcinoma: Therapeutic Potential of Atractylodes lancea (Thunb.) DC.' and I propose the

acceptance of the article.

Reviewer #2: The authors have significantly improved this manuscript since the last submission. All my concerns have been addressed and I recommend publication.

7. PLOS authors have the option to publish the peer review history of their article (what does this mean? ). If published, this will include your full peer review and any attached files.

**Do you want your identity to be public for this peer review?** For information about this choice, including consent withdrawal, please see our Privacy Policy .

Reviewer #1: **Yes: ** mohamed abdelkarim

Reviewer #2: **Yes: ** Willem Buys

---

## [Editor Report · Acceptance letter]

PONE-D-24-55645R2

PLOS ONE

Dear Dr. Na-Bangchang,

I'm pleased to inform you that your manuscript has been deemed suitable for publication in PLOS ONE. Congratulations! Your manuscript is now being handed over to our production team.

Kind regards,

on behalf of

Dr. Keun-Yeong Jeong

Academic Editor

PLOS ONE